# Allelic Variants in Established Hypopituitarism Genes Expand Our Knowledge of the Phenotypic Spectrum

**DOI:** 10.3390/genes12081128

**Published:** 2021-07-25

**Authors:** Marilena Nakaguma, Nathalia Garcia Bianchi Pereira Ferreira, Anna Flavia Figueredo Benedetti, Mariana Cotarelli Madi, Juliana Moreira Silva, Jun Z. Li, Qianyi Ma, Ayse Bilge Ozel, Qing Fang, Amanda de Moraes Narcizo, Laís Cavalca Cardoso, Luciana Ribeiro Montenegro, Mariana Ferreira de Assis Funari, Mirian Yumie Nishi, Ivo Jorge Prado Arnhold, Alexander Augusto de Lima Jorge, Berenice Bilharinho de Mendonca, Sally Ann Camper, Luciani R. Carvalho

**Affiliations:** 1Developmental Endocrinology Unit, Laboratory of Hormones and Molecular Genetics LIM/42, Division of Endocrinology, Hospital das Clinicas da Faculdade de Medicina da Universidade de São Paulo (FMUSP), Av Dr Eneas de Carvalho Aguiar, 155, 2 Andar, Bloco 6, São Paulo 05403-000, Brazil; mari_nakaguma@yahoo.com.br (M.N.); nathalia.pferreira2@gmail.com (N.G.B.P.F.); anna.detti@gmail.com (A.F.F.B.); marianacmadi89@gmail.com (M.C.M.); julianamoreira@usp.br (J.M.S.); lucianam@usp.br (L.R.M.); marianafunari@usp.br (M.F.d.A.F.); minishi@usp.br (M.Y.N.); iarnhold@usp.br (I.J.P.A.); beremen@usp.br (B.B.d.M.); 2Laboratorio de Sequenciamento em Larga Escala (SELA), Faculdade de Medicina FMUSP, Universidade de São Paulo, São Paulo 01246-903, Brazil; amnarcizo@usp.br (A.d.M.N.); lais.cavalca@fm.usp.br (L.C.C.); 3Department of Human Genetics, University of Michigan, Ann Arbor, MI 48109-5618, USA; junzli@med.umich.edu (J.Z.L.); qzm@umich.edu (Q.M.); aozel@umich.edu (A.B.O.); qing.fang@regeneron.com (Q.F.); 4Genetic Endocrinology Unit (LIM25), Division of Endocrinology, Hospital das Clinicas da Faculdade de Medicina da Universidade de São Paulo (FMUSP), São Paulo 01246-903, Brazil; alexj@usp.br

**Keywords:** GH1, SOX3, TGIF1, hypopituitarism, allelic variants

## Abstract

We report four allelic variants (three novel) in three genes previously established as causal for hypopituitarism or related disorders. A novel homozygous variant in the growth hormone gene, *GH1* c.171delT (p.Phe 57Leufs*43), was found in a male patient with severe isolated growth hormone deficiency (IGHD) born to consanguineous parents. A hemizygous *SOX3* allelic variant (p.Met304Ile) was found in a male patient with IGHD and hypoplastic anterior pituitary. YASARA, a tool to evaluate protein stability, suggests that p.Met304Ile destabilizes the SOX3 protein (ΔΔG = 2.49 kcal/mol). A rare, heterozygous missense variant in the TALE homeobox protein gene, *TGIF1* (c.268C>T:p.Arg90Cys) was found in a patient with combined pituitary hormone deficiency (CPHD), diabetes insipidus, and syndromic features of holoprosencephaly (HPE). This variant was previously reported in a patient with severe holoprosencephaly and shown to affect TGIF1 function. A novel heterozygous *TGIF1* variant (c.82T>C:p.Ser28Pro) was identified in a patient with CPHD, pituitary aplasia and ectopic posterior lobe. Both *TGIF1* variants have an autosomal dominant pattern of inheritance with incomplete penetrance. In conclusion, we have found allelic variants in three genes in hypopituitarism patients. We discuss these variants and associated patient phenotypes in relation to previously reported variants in these genes, expanding our knowledge of the phenotypic spectrum in patient populations.

## 1. Introduction

Congenital hypopituitarism is a rare disorder with a prevalence of 1/3000 to 1/4000 births, characterized by deficient production of one or more pituitary hormones [1]. Clinical manifestations are variable. Pituitary hormone deficiency can occur with or without syndromic features, manifest early at birth or during infancy, and progress with age [2,3,4].

Genetic investigation is fundamental to understand pituitary development and to allow early diagnosis, predict disease progression, and offer genetic counseling. Early patient studies used Sanger sequencing of candidate genes, such as transcription factors that were critical for pituitary development during embryogenesis in mice [3].

In the last three decades, pathogenic allelic variants in more than 30 genes were recognized as a cause of congenital hypopituitarism [3]. The application of massively parallel sequencing, in targeted gene panels, exomes or whole genomes, has made it possible to identify new genes and rare variants involved in pituitary development and disease and to expand the phenotype associated with previously known genes [5,6,7,8,9,10,11,12,13,14]. Genotype–phenotype correlations are still difficult to discern, given the variability of features among patients with lesions in the same gene.

In this paper we describe variants in *GH1*, *SOX3* and *TGIF1*, three genes that are already associated with hypopituitarism. These variants were identified by exome sequencing or by the sequencing a panel of selected genes in a large cohort of patients with combined pituitary hormone deficiency ascertained in a single Brazilian medical center.

## 2. Materials and Methods

### 2.1. Ethical Procedures

All patients or their parents gave their permission to take part in the present study, which was approved by the Brazilian national ethical committee under the number CAAE 0642812.4.0000.0068

### 2.2. Patients

The patients described here in detail are under the care of the endocrinology clinic at the Hospital das Clinicas, University of São Paulo Medical School.

### 2.3. DNA Extraction

DNA was extracted from peripheral blood samples using salting out method [15]. The panel for capturing exonic regions of known hypopituitarism genes was designed using SureDesign tool (Agilent, Santa Clara, CA, USA), and DNA was sequenced using Illumina platform NextSeq 550 at SELA (Sao Paulo, SP, Brazil). Libraries for exome sequencing were prepared using either NimbleGen v3 (Roche, Basel, Switzerland) or SureSelect v6 (Agilent, Santa Clara, CA, USA). Sequencing was performed at the Sequencing Core at Michigan University (Ann Arbor, MI, USA) or SELA, respectively, using the Illumina platform HiSeq 2000.

Sequencing quality was checked using FASTQC (https://www.bioinformatics.babraham.ac.uk/projects/fastqc/, accessed on date 31 May 2019), and BWA-MEM was used to assemble the sequence to the reference genome hg19 (http://bio-bwa.sourceforge.net/bwa.shtml, accessed on date 31 May 2019). Variants were called with either GATK [16] or FreeBayes [17] and annotated with Annovar [18].

### 2.4. Filtering Process and Sequencing Analysis

Exome and panel variant sequencing analysis were similar, as they were performed on genomic DNA samples from individual patients. The filtering pipeline took into consideration variants identified in exonic and splice site regions that were present with a Minor Allele Frequency (MAF) of less than 1% in international and national population databases: gnomAD v3.1.1 (gnomad.broadinstitute.org, accessed on date 6 March 2021), 1000 Genomes, ABraOM (abraom.ib.usp.br, accessed on date 6 March 2021) [19] and internal database SELAdb (intranet.fm.usp.br/sela, accessed on date 6 March 2021) [20]. First, homozygous or compound heterozygous variants were considered assuming an autosomal recessive disorder. If no variants of interest were evident, the search was expanded to consider heterozygous variants. Algorithms such as MutationTaster v2., MutationAssessor, SIFT, PolyPhen2, and Human Splicing Finder were used to predict which variants would be deleterious. Variants were then classified according to recommendations of the American College of Medical Genetics (ACMG/AMP) with the help of Varsome (varsome.org, accessed on date 8 March 2021).

### 2.5. Bioinformatics Tools to Check Allelic Variant Impact

We used the RNAfold server from ViennaRNA Web Services to predict mRNA secondary structure (http://rna.tbi.univie.ac.at/cgi-bin/RNAWebSuite/RNAfold.cgi, accessed on date 15 August 2020). We determined RNA base pair probability and optimal folding for *SOX3* wild type mRNA (Met304), a rare variant reported in gnomAD (Met304Val), and the candidate variant (Met304Ile).

Protein stability was calculated using YASARA (http://yasara.org, accessed on date 15 August 2020) via the FoldX plugin (http://foldxsuite.crg.eu/, accessed on date 15 August 2020). A variant was considered destabilizing when ΔΔG was positive, taking into consideration the tool’s error margin of ΔΔG = ±0.5 kcal/mol.

## 3. Results

### 3.1. Patient 1 with Allelic Variant GH1 c.171delT, p.Phe 57Leufs*43, chr17:61995706:delA

#### 3.1.1. Clinical, Laboratory, and Image Features

A male patient, age 9.5 years, presented at his first visit with a height of 87 cm (−7.65 SD) and delayed bone age by 4.6 years (Table 1). A clonidine stimulation test confirmed growth hormone (GH) deficiency with a GH peak < 0.25 ng/dL which, for the radioimmunoassay method, was considered unresponsive in cases where GH < 7 ng/dL. A good response was obtained in the first year of treatment with somatotropin, with a growth rate of 16.9 cm/year and a delta Z-score of 2.18 (height at the end of first year −5.47 SD) (Table 1).

Magnetic resonance imaging (MRI) revealed a normal anterior pituitary lobe, a visualizable stalk, and appropriately positioned neurohypophysis (Table 2).

Puberty occurred spontaneously at age 14 years, and a pubertal block was administered from 14 years and 8 months to 16 years and 7 months. His final height was 170.5 cm (−0.63 SD) (Table 1). The patient had bilateral cryptorchidism, which was surgically corrected at age 12. At the age of 14, he developed hypergonadotrophic hypogonadism. As an adult he underwent unsuccessful assisted reproduction.

#### 3.1.2. Molecular Results

The parents of the proband, II.3, were first cousins and were unaffected (Figure 1). The male proband had two older sisters and a younger sister. He was homozygous for the allelic variant *GH1* c.171delT; p.Phe 57Leufs*43;chr17:61995706:delA. Both his sister (II.4) and mother (I.1) were heterozygous for the variant, and they had no abnormal features. Another sister (II.2) died at 5 years of age and had a phenotype suggestive of growth hormone deficiency (GHD), including a saddle nose, frontal bossing, and short stature, but no DNA was available for testing.

The *GH1* variant c.171delT (p.Phe57Leufs*43) has never been described in association with hypopituitarism, either in OMIM or in Genecards. This variant is absent in population databases, including Exome Aggregation Consortium (EXAC), gnomAD, and the Brazilian population databases (SELA and ABraOM) (Table 3). The variant *GH1* c.171delT (p.Phe57Leufs*43) was visually confirmed using integrated genome viewer (IGV), and it is classified as pathogenic by Varsome [21].

### 3.2. Patient 2 with Allelic Variant in SOX3 (c.912G>A;p.Met304Ile; chrX:139586314:C>T)

#### 3.2.1. Clinical, Laboratory and Image Features

A male patient, the son of non-consanguineous parents, was born at term, with appropriate weight: 3300 g (−0.73 SDS). There were no perinatal complications, and his neurological development was normal. Short stature was noticed at 2 years of age, and at 6 years of age he was diagnosed with growth hormone deficiency. Somatotropin treatment began at age 7 with an initial height of 95 cm (−4.67 SDS) and was continued to age 15.

Spontaneous puberty occurred at age 13 and was blocked from 13 years and 9 months to 14 years and 8 months. At 16 years and 6 months, his bone age was 16 years, and his final height was 153 cm (−2.9 SDS) (Table 1). IGF1 was 200 ng/mL (NV −227–964 ng/mL) and in the insulin tolerance stimulation test (ITT), glycemia trough was 32 mg/dL and maximum peak of GH 0.9 ng/mL. Only growth hormone deficiency was confirmed, and somatotropin 1 U/day was reintroduced. MRI revealed pituitary hypoplasia and ectopic neurohypophysis located at the level of the optic chiasm (Table 2).

#### 3.2.2. Molecular Results

Using targeted gene panel sequencing, a hemizygous variant in the *SOX3* gene was found in a male patient: c.912G>A;p.Met304Ile; chrX:139586314:C>T. This variant has been reported in gnomAD in two hemizygotes, with a population frequency of 0.07e3, and it is predicted to be deleterious by SIFT, MutationTaster, MutationAssessor and PolyPhen2 (Table 3). The patient’s mother, father, brother, sister and maternal uncle were all phenotypically normal and were screened for the variant. Both the unaffected mother and sister were carriers. In many X-linked diseases, female carriers are unaffected due to preferential inactivation of the mutant X-chromosome. The father, brother and uncle were negative for the variant. (Figure 2A).

The p.Met304Ile variant is located just outside the SOXp region, which is a highly conserved region in the SOX family of proteins (Figure 2B,C). Two in silico studies were conducted to assess variant pathogenicity. The c.912G>A substitution is predicted to cause loss of a hairpin in the mRNA secondary structure, although the significance of such a change is unclear (Figure 3). The p.Met304Ile variant is predicted to be destabilizing (ΔΔG = 2.49 kcal/mol) for the protein based on in silico analysis with the YASARA tool. For comparison, the previously reported missense variants p.Ser150Tyr and p.Pro142T were tested with the same tool and found to be destabilizing also (ΔΔG = 5.75 kcal/mol and 5.85 kcal/mol, respectively). By contrast, the variant alleles p.Arg5Gln (196 hemizygotes, allele frequency 0.003194 in gnomAD) and p.Met304Val (1 hemizygote, allele frequency 0.00004660 in gnomAD) are predicted to be tolerated as the ΔΔG values (p.Arg5Gln ΔΔG = 0.83 kcal/mol and p.Met304Val 0.47 kcal/mol) are within the tool’s error margin of ΔΔG = ±0.5 kcal/mol.

### 3.3. Patient 3 with Allelic Allelic Variant TGIF1 (c.268C>T;p.Arg90Cys; chr18:3457387:C>T)

#### 3.3.1. Clinical Features and Test Results

A female patient, II.2, was born to non-consanguineous parents and delivered by caesarean section at 37 weeks. Her twin sister was diagnosed with holoprosencephaly (HPE) and died at birth, II.3.

The patient was born small for gestational age: 2505 g (−0.75 SDS), 46 cm (−0.63 SDS) and head circumference 32 cm (−0.65). The patient presented with serious complications at birth, including prolonged jaundice, hypothermia, hyponatremia and seizures (Table 1). She also had syndromic features that included craniofacial malformation, hypertelorism, and nystagmus. She underwent surgical correction of her cleft palate on her fifthday of life. She was severely affected with significant neuropsychomotor developmental delay and required enteral feeding. At her second month of life, she was diagnosed with congenital hypopituitarism and started replacement therapy with prednisolone at 0.6 mg per day, levothyroxine at 12.5 mcg per day, and desmopressin at 0.012 mg per day. Recombinant growth hormone replacement was started when she was 2 years old, and spontaneous menarche occurred when she was 12 years old. Magnetic resonance imaging (MRI) (40 days) revealed absence of septum pellucidum, semilobar holoprosencephaly with partial fusion of thalamus and basal ganglia, dysgenesis of the corpus callosum, small third ventricle, fusion of frontal lobe, wide communication of the lateral ventricle, rudimentary horns and ectopic posterior pituitary (Table 2).

#### 3.3.2. Molecular Results

Using whole exome sequencing, we identified an allelic variant in *TGIF1* (c.268C>T;p.Arg90Cys; chr18:3457387:C>T) (Figure 4A) classified by the American College of Medical Genetics (ACMG) and Association for Molecular Pathology (AMP) as likely pathogenic. This variant is absent in ExAC and gnomAD, as well as in the Brazilian population (SELA and ABRAOM) (Table 3). The presence of this variant was confirmed in the patient and in her unaffected father and sister (Figure 4C). The arginine at position 90 is well conserved among species (Figure 4E).

### 3.4. Patient 4 with Allelic Variant TGIF1 (c.82T>C;p.Ser28Pro; chr18:3456417:T>C)

#### 3.4.1. Clinical Features and Test Results

A male patient was born at term to non-consanguineous parents. He weighed 3850 g (+1.79 SDS) and was 48 cm long (−0.55 SDS). His neuropsychomotor development was normal. He presented at 4.9 years with a height of 86.7 cm (−4.5 SDS) and a bone age of 2.5 years. A clonidine stimulation test was performed, and the maximum GH response was 0.4 ng/mL. He was given an insulin tolerance stimulation test (ITT) at 8.6 years, and the GH peak was 0.1 ng/mL and cortisol was 7.2 µg/dL (basal of 8.0 µg/dL). This confirmed the presence of GH and ACTH deficiencies (Table 1). He received rGH replacement from 5 to 19 years. His growth velocity was 12 cm/year in the first year of treatment, and his final height was 168.5 cm (SDS-0.62). He presented a baseline cortisol of 6.0 µg/dL at 9.1 years and started treatment with hydrocortisone acetate. Puberty was induced with testosterone cypionate when he was 14.4 years of age (Table 1). MRI (5 year) revealed pituitary aplasia, interrupted pituitary stalk, and ectopic posterior lobe (Table 2).

#### 3.4.2. Molecular Results

A heterozygous *TGIF1* c.82T>C;p.Ser28Pro; chr18:3456417:T>C variant was identified with targeted gene panel sequencing. This is classified as a variant of uncertain significance according to ACMG/AMP. The variant is absent in ExAC, gnomAD, and the Brazilian population databases (SELA and ABraOM) (Table 3). His unaffected mother and half-brother also are heterozygous for this variant (Figure 4B). The serine at position 28 is well conserved among species (Figure 4D).

## 4. Discussion

We identified variants in three hypopituitarism genes in four Brazilian patients using next generation sequencing.

### 4.1. GH1 Gene

*GH1* was the first gene recognized as a monogenic cause of isolated growth hormone deficiency (IGHD) in 1981 [22]. The gene encoding *GH1* is located on the long arm of chromosome 17 (17q22–24) in a cluster of five related genes, including two chorionic somatotropin genes *CHS1* and *CHS2*, the *CSHP1* pseudogene and *GH2*, which is a variant of growth hormone expressed in the placenta. *GH1* consists of five exons and four introns, and the primary protein product is 22 kDa [23]. IGHD is classified in four subcategories: autosomal recessive (type IA and IB), autosomal dominant (type II) and X-linked (type III). Type IB is a rare form of IGHD (2%), featuring short stature, low serum GH concentrations and good response to treatment with rhGH, without formation of antibodies. It is more frequent in consanguineous families, and *GH1* mutations can be frameshift, missense, homozygous nonsense, or splice site mutations in *GH1* [24]. This patient was classified as type IB (MIM # 612781) due to his clinical characteristics, good response to treatment with recombinant human GH (rhGH), and the likelihood that the early frameshift creates a loss of function. Therefore, the allelic variant that we report *GH1* c.171delT (p.Phe 57Leufs*43) is a new, pathogenic variant.

### 4.2. SOX3 Gene

A variety of gain and loss of function mutations have been identified in *SOX3*, including gene duplication, deletion, alanine tract expansion, and missense variants. The patient phenotypes are variable, and they can include intellectual disability (MIM # 300123), midline and forebrain abnormalities, isolated growth hormone deficiency, or combined pituitary hormone deficiencies (MIM # 312000) [25].

Two previously reported missense variants in SOX3, p.Ser150Tyr [26] and p.Pro142Thr [27], are located in the N-terminal tail of the HMG (High Mobility Group) domain, and the patients presented with a complex phenotype of syndromic, combined pituitary hormone deficiency. Cell culture studies demonstrated that the p.Pro142T variant increases SOX3-mediated transcriptional activation of *HESX1* and diminishes repression of β-catenin-mediated transcription [27]. Although no functional studies are reported for the p.Ser150Tyr variant, the inheritance pattern is consistent with pathogenicity, as three affected brothers were hemizygous and multiple carrier females were unaffected. The lack of effect in females may be explained by preferential inactivation of the abnormal X chromosome [28].

The SOX3 p.Met304Ile variant that we identified is located just outside the SOXp domain, which is a highly conserved domain ending in codon 302. Variant segregation in the family conforms to expectations, as only the patient carried the variant in a hemizygous state. His mother and sister were unaffected carriers. Protein stability prediction tools are consistent with a destabilizing effect of this variant and two reported missense variants. According to ACMG, the p.Met304Ile variant is classified as a VUS as there is insufficient evidence in favor of pathogenicity. Recently, this variant was reported in two hemizygotes in TOPMed (ss3623368805) and the newest version of gnomAD, with a frequency of 0.07e3. Based on the conflicting data about pathogenicity of the SOX3 change (p.Met304Ile), the clinical relevance of the variant remains still unclea. Since molecular diagnosis has been reached out by sequencing of a target gene panel only WES or WGS approaches are required to exclude alternative molecular events underlying the disorder

### 4.3. TGIF1 Gene

We identified two *TGIF1* variants that were absent in ExAC and gnomAD, as well as in the Brazilian population databases (SELA and ABraOM). *TGIF1* (c.268C>T:p.Arg90Cys) was identified in the present study by whole exome sequencing in a patient with features of HPE (MIM # 142946) and combined pituitary hormone deficiency. The p.Arg90Cys variant was previously reported as a de novo mutation in a 22 wk fetus with alobar HPE, brachycephaly, hypotelorism, median orofacial cleft and nasal hypoplasia [29]. The pituitary hormone status of this fetus was not reported, but pituitary hormone deficiencies are present in 63% of non-chromosomal, non-syndromic HPE patients [30]. Later, functional studies confirmed that the p.Arg90Cys variant abolishes binding to the TGIF consensus site, reduces the repressive properties of TGIF1 by affecting interaction with SMAD3 and RXR [31]. Thus, this variant is pathogenic.

We identified a new *TGIF1* variant (c.82T>C:p.Ser28Pro) in a patient evaluated by our targeted gene panel. The patient had LH, GH and ACTH deficiencies, pituitary aplasia, interrupted pituitary stalk, and ectopic posterior lobe but no major cerebral malformations or features of HPE. This variant affects the same codon as a previously reported missense mutation (c.83C>T:pSer28Cys) found in a patient with hypotelorism, congenital nasal pyriform aperture stenosis, single central incisor, agenesis of the corpus callosum, microcephaly and developmental delay [32,33]. This TGIF region contains a conserved motif (PLDLS) with an important transcriptional repression activity. Previous functional studies demonstrated that p.Ser28Cys results in decreased RXR and TGFβ dependent transcriptional repression and loss of CtBP interaction [31]. This variant is reported in TOPMed (rs1171035105, MAF = 1/125568) as a VUS, and although this variant is classified as variant of uncertain significance (ACMG/AMP), it seems plausible that it is pathogenic.

Both The p.Arg90Cys and p.Ser28Pro *TGF1* variants were identified in healthy members of the families. However, it is common for variants in genes that cause HPE to exhibit incomplete penetrance and variable expressivity.

Tatsi et al. reported a female patient with solitary central incisor, low GH, TSH and gonadotropins, adenohypophysis hypoplasia, absence of the pituitary stalk and ectopic posterior pituitary lobe but no HPE brain defects. The patient and her asymptomatic father carried a heterozygous c.799C>T, p.Q267X TGIF1 variant, predicting truncation of TGIF1 and loss of the last 5 amino acids [34].

To the best of our knowledge, the patient reported here is the first one with CPHD and a *TGIF1* variant without HPE or craniofacial midline defects.

## 5. Conclusions

In conclusion, we have found four allelic variants in three genes in hypopituitarism patients. It is clear that variants in *SOX3* and *TGIF1* produce variable and incompletely penetrant features.

## Figures and Tables

**Figure 1 genes-12-01128-f001:**
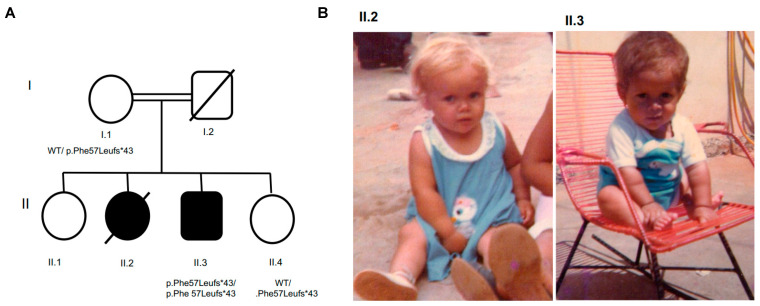
Pedigree and inheritance of the *GH1* c.171delT (p.Phe 57Leufs*43) allelic variant (**A**). Pedigree of the proband II.3 indicating recessive inheritance of p.Phe57Leufs*43 (**B**). Photographs (obtained with permission) of the proband (II.3, far right) and his sister (II.2) with features of growth hormone deficiency.

**Figure 2 genes-12-01128-f002:**
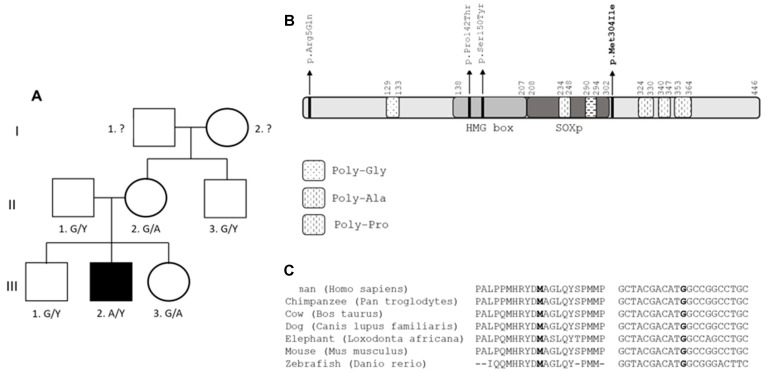
Segregation and conservation of SOX3 variant p.Met304Ile. (**A**) Family pedigree of the proband, III.2, illustrating the segregation of the G to A variant in SOX3 that results in p.Met304Ile. The male proband was the only affected individual, (III.2). Letters below each family member represent the genotype, considering G to be the normal allele and A the variant. (**B**) Protein diagram for SOX3. The missense variants mentioned in text are indicated. The high mobility group DNA binding domain (HMG) and SOXp domains are highlighted, as well as portion of the protein containing amino acid repeats. (**C**) SOX3 protein (**left**) and cDNA (**right**) are highly conserved across species at and around p.Met304. 3. The amino acid Met (M) and nucleotide (G) that are mutated in p.Met304Ile are in bold.

**Figure 3 genes-12-01128-f003:**
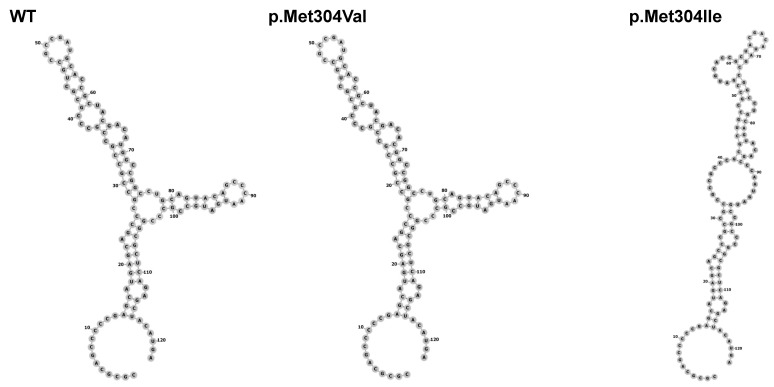
SOX3 mRNA structure change. While the p.Met304Val variant maintains the same structure as the wild type (WT) variant, the p.Met304Ile, present in the patient, loses a hairpin formation in the mRNA.

**Figure 4 genes-12-01128-f004:**
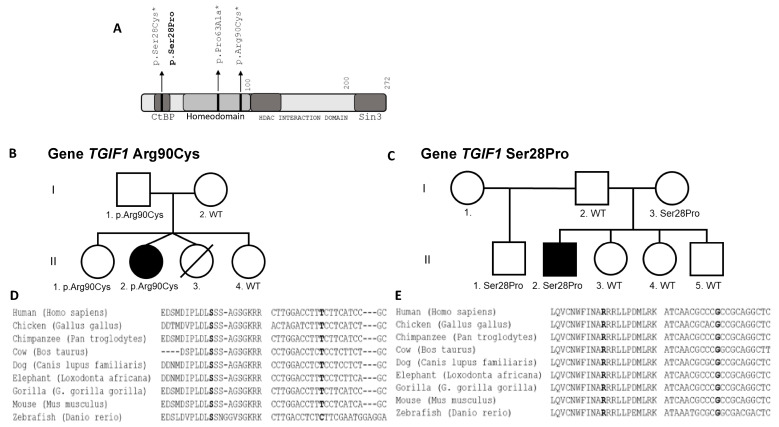
Segregation of *TGIF1* variants and evolutionary conservation. (**A**) Human missense variants in TGIF1 are indicated in the protein diagram. Two variants are in the CtBP domain of TGIF1 which normally interacts with the carboxy terminus binding protein to repress expression of target genes. TGIF1 is a member of the three amino acid loop extension (TALE) homeodomain family of genes, and two variants are in this DNA binding domain. The variants in bold are from the present study, and the other two variants were previously shown to be deleterious in functional studies. The histone deacetylase (HDAC) interaction domain and Sin3a domains of TGIF1 are also important for repression. (**B**) Family pedigree showing segregation of TGIF1 p.Arg90Cys variant present in the heterozygous state in the proband, II.2, and her unaffected father and sister, I.1 and II.1, respectively. Her mother and sister, I.2 and II.4, are homozygous for the normal allele. (**C**) Family pedigree showing segregation of *TGIF1* p.Ser28Pro variant present in a heterozygous state in the proband, II.2 and his unaffected mother and half-brother. The father, I.2, and the siblings, II.3, II.4 and II.5 are homozygous for the normal allele. (**D**) TGIF1 protein (**left**) and cDNA (**right**) are evolutionarily conserved in the region around Ser28. The amino acid and nucleotide mutated in p.Ser28Pro are in bold. (**E**) TGIF1 protein (**left**) and cDNA (**right**) are evolutionarily conserved in the region around Arg90. The amino acid and nucleotide mutated in p.Arg90Cys are in bold.

**Table 1 genes-12-01128-t001:** Phenotype and endocrine investigations of patients.

Patient	Age at Testing in Years	Initial Height SDS	Puberty I/S (Years)	Final Height SDS	Target Height SDS	GH Peak μg/L	Cortisol Peak n·mol/L (NR > 550)	FT4 p·mol/L (NR)	TSH mU/L (NR)	IGF1 ng/mL (NR)	IGFBP3 mg/L (NR)	PRL mU/L (NR)
1	9.5	−7.65	S (14)	−0.63	−0.63	<0.25	NA	NA	6.0 (0.5–4.4)	NA	NA	54
2	6	−4.67	S (13)	−3.2	−0.7	0.9	552	0.97 (0.7–1.5)	2.37 (0.5–4.4)	200 (227–964)	4.2 (3.3–5.7)	340 (<450)
3	0.66	−4.8	-	Still growing	+0.55	0.15 *	39	**	6.3 (0–20)	25 (48-313)	NA	278 (57–717)
4	4.9	−4.55	I (15)	−0.93	0.34	0.4	NA	***	3.11 (0.5–4.2)	<18 (25–68)	0.4 (1.5–3.4)	42.5 (42-170)

Induced—I, Spontaneous—S, SDS—standard deviation score, NR—normal range, NA—not available. * Basal during hypoglycemia of 27 mg/dL, ** Total T4–6.68 RV 4.5–22.2, *** Total T4 10.2 (7.7–49.8), GH cut off > 3.3 mcg/L (IFMA).

**Table 2 genes-12-01128-t002:** Molecular diagnosis and patients’ clinical and image features.

Patient	M/F	Gene	Allelic Variant	Inheritance	Hormone Deficiencies	MRI (CA)
1	M	GH1	p.Phe57Leufs*43	AR	IGHD HyperHypogon	TPP normal AP 7.5 year
2	M	SOX3	p.Met304Ile	X-linked	IGHD	EPP AP aplasia 8 year
3	F	TGIF1	p.Arg90Cys	AD–IC	GH, TSH, ACTH, PRL and ADH	HPE 40 days
4	M	TGIF1	p.Ser28Pro	AD–IC	GH, TSH, ACTH, LH/FSH, PRL	EPP AP aplasia 5.1 year

M/F, male/female; MRI, magnetic resonance imaging; AR, autosomal recessive; AD, autosomal dominant; IC, Incomplete penetrance; IGHD, Isolated growth hormone deficiency; HyperHypogon, hypergonadotropic hypogonadism; DI, diabetes insipidus; AP, anterior pituitary; EPP, ectopic posterior pituitary; TPP, topic posterior pituitary; holoprosencephaly (HPE).

**Table 3 genes-12-01128-t003:** Allelic variant classification according to ACMG.

Gene	Variant	OMIM/Genecards	gnomAD	ABraOM	SELAdb	ACMG	CADD	REVEL
*GH1*	NM_000515.5:c.171delT; p.Phe57Leufs*43 (chr17:61995706: delA)	Never related to hypopituitarism	Absent	Absent	Absent	Pathogenic	22.4	No data
*SOX3*	NM_005634.3:c.912G>A;p.Met304Ile (chrX:139586314: C>T)	Never related to hypopituitarism	Absent	Absent	Absent	VUS	23.4	0.670
*TGIF1*	NM_173208.1 c.268C>T: p.Arg90Cys (chr18:3457387: C>T)	Never related to hypopituitarism	Absent	Absent	Absent	Likely pathogenic	28.8	0.9729
*TGIF1*	NM_173208.1 c.82T>C; p.Ser28Pro (chr18:3456417: T>C)	Never related to hypopituitarism	Absent	Absent	Absent	VUS	22.8	0.279

VUS variant of unknown significance.

## Data Availability

Data is available at the Sequence Read Archive (SRA: https://www.ncbi.nlm.nih.gov/sra, accessed on 4 July 2021) under SUB10027627.

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
