# Peer review of "Allelic Variants in Established Hypopituitarism Genes Expand Our Knowledge of the Phenotypic Spectrum"

_genes, 2021, doi:10.3390/genes12081128_

Round 1

Reviewer 1 Report

The manuscript by Nakaguma et al. provides new data regarding four rare variants affecting three genes already associated to hypopituitarism. In this context, the authors report 4 patients with combined pituitary hormone deficiency, aiming to expand the relative phenotypes. The manuscript is well written and explains the scope of author's investigation. However, there are still missing data to warrant a satisfactory level of confidence in assigning pathogenicity to the variants identified in the study. In addition, one out of four variant the hypothesis of pathogenicity is not supported by data of allele frequency provided by population studies. There are reported data supporting evidence that the genetic basis for combined pituitary hormone deficiency is complex, characterized by complex etiology, and involving at least 30 genes and some rare copy number variations (as reported by following studies PMID:27828722 and PMID:32796691). Taking into the account that at least 3 variants identified in this study are indicated as new, more functional validations (apart predictions by YASARA tool), or a dedicated study, are required to demonstrate the biologic relevance and clinical significance of these new variants.

In detail:

  1. RESULTS, Patient 1 with allelic variant GH1 c.171delT, p.Phe57Leufs*43. The diagnostic hypothesis for this family is likely be supported the clinical plausibility, segregation data, and by the fact that there is no frequency of the variant in the general population. However, following the recommendation for genetic assessments in families with consanguineous elements, the SNP array revealing long contiguous regions of homozygosity (ROH) is strongly required to exclude the occurrence of structural variants or co-occurrence of genetic modifiers. Please provide these data, if available.

  1. RESULTS, Patient 2 with allelic variant in SOX3 (c.912G>A;p.Met304Ile). The event found in a sporadic case is a VUS, annotated as rare. It has also been reported in population database (rs775380773, gnomAD v3.1.1) with a maxAF=0.07e3, and a total of 8 alleles with 2 hemizygosity have also been identified in general population. These data do not support the occurrence of SOX3 variant as the unique event underlying the condition. It is very curious how the same variant is predicted by YASARA tool to be destabilizing (ΔΔG=2.49 kcal/mol) for the protein. This review suggests the authors investigate further on it, since it could contribute to the molecular pathomechanism of the disorder. Finally, as a target gene panel sequencing has been performed in this family, this reviewer also suggests looking for alternative diagnostic hypotheses by using WES/WGS approaches.

  1. RESULTS, Patient 3 with allelic allelic variant TGIF1 (c.268C>T;p.Arg90Cys).

  1. - The variant corresponds to the 3:c.310C>T, NP_775299.1:p.Arg104Cys. On note, please provide the NM accession number for all variants identified in the present study. 
  2. - The variant has previously been identified (rs1555650923, with no MAF), reported in ClinVar database, and previously been described as de novo event in two unrelated families with HPE, type 4 (PMID: 11810641; PMID: 16962354). As the authors indicated, although the patient 3 shows clinical and instrumental features of HPE, it never has been associated to hypopituitarism. However, available clinical data in previously reported cases were not complete and we cannot exclude a priori that the growth deficiency and other features documented in those subjects cannot be associated also to hypopituitarism.
  3. -The 268C>T;p.Arg90Cys variant has been found to be shared by other healthy members of the family (the healthy sister and father). This review agrees that clinical spectrum of malformations is consistent with the fact that the impact of TGIF1 variants may be variable and incomplete penetrance can also occur. In this context, a dedicated study on the primary transcript, or CpG islands within the promoter, may help to better understand this issue.

  1. RESULTS, Patient 4 with allelic variant TGIF1 (c.82T>C;p.Ser28Pro). The variant has previously been identified (rs1171035105, MAF=1/125568, TOPMED). There is a large number of isoforms for this gene/protein, please provide the relative NM.. (or at least ENST..) and NP.. numbers. The variant is classified as VUS but deserves to be reported by this work, corroborating what has been previously reported on the affected Serine residue, thus providing clinical relevance for HPE. It could be interesting obtain a clinical comparison between the clinical profile of the subject carrying TGIF1(ENST00000330513.5):c.470C>G, p.Ser157Cys (rs121909066, PMID: 10835638) and the patient 4 of the present study.

  1. Table 3. Please insert the in silico predictions referring each variant (i.e. CADD and/or REVEL).

  1. DISCUSSION: the authors should add the MIM reference numbers (i.e. Growth hormone deficiency, isolated, type IA, MIM#262400).

Overall, the manuscript provides evidence that combined pituitary hormone deficiency may be caused by different molecular events, that can be identified by high-throughput sequencing approaches. .

Author Response

The manuscript by Nakaguma et al. provides new data regarding four rare variants affecting three genes already associated to hypopituitarism. In this context, the authors report 4 patients with combined pituitary hormone deficiency, aiming to expand the relative phenotypes. The manuscript is well written and explains the scope of author's investigation. However, there are still missing data to warrant a satisfactory level of confidence in assigning pathogenicity to the variants identified in the study. In addition, one out of four variant the hypothesis of pathogenicity is not supported by data of allele frequency provided by population studies. There are reported data supporting evidence that the genetic basis for combined pituitary hormone deficiency is complex, characterized by complex etiology, and involving at least 30 genes and some rare copy number variations (as reported by following studies PMID:27828722 and PMID:32796691). Taking into the account that at least 3 variants identified in this study are indicated as new, more functional validations (apart predictions by YASARA tool), or a dedicated study, are required to demonstrate the biologic relevance and clinical significance of these new variants.

In detail:

  1. RESULTS, Patient 1 with allelic variant GH1 c.171delT, p.Phe57Leufs*43. The diagnostic hypothesis for this family is likely be supported the clinical plausibility, segregation data, and by the fact that there is no frequency of the variant in the general population. However, following the recommendation for genetic assessments in families with consanguineous elements, the SNP array revealing long contiguous regions of homozygosity (ROH) is strongly required to exclude the occurrence of structural variants or co-occurrence of genetic modifiers. Please provide these data, if available.

Answers: Thank you for your comments.  We do not have SNP data.

However, the clinical phenotype expected for a homozygous loss of function mutation in GH1 is consistent with the phenotype of this patient.  Thus, co-occurrence of genetic modifiers is unlikely.

RESULTS, Patient 2 with allelic variant in SOX3 (c.912G>A;p.Met304Ile). The event found in a sporadic case is a VUS, annotated as rare. It has also been reported in population database (rs775380773, gnomAD v3.1.1) with a maxAF=0.07e3, and a total of 8 alleles with 2 hemizygosity have also been identified in general population. These data do not support the occurrence of SOX3 variant as the unique event underlying the condition. It is very curious how the same variant is predicted by YASARA tool to be destabilizing (ΔΔG=2.49 kcal/mol) for the protein. This review suggests the authors investigate further on it, since it could contribute to the molecular pathomechanism of the disorder. Finally, as a target gene panel sequencing has been performed in this family, this reviewer also suggests looking for alternative diagnostic hypotheses by using WES/WGS approaches.

Answer: The text has been updated to reflect the frequency of this allele based on the newest version of gnomAD in RESULTS line 175-176 and in DISCUSSION lines 311-315.  Additional testing is required to determine whether the p.Met304Ile variant affects SOX3 transactivation properties.  We agree that WES or WGS may reveal disease causing variants in genes not included in our panel.  We have modified the text accordingly.

RESULTS, Patient 3 with allelic allelic variant TGIF1 (c.268C>T;p.Arg90Cys).

  1. - The variant corresponds to the 3:c.310C>T, NP_775299.1:p.Arg104Cys. On note, please provide the NM accession number for all variants identified in the present study. 

Answer: NM accession number for all variants identified in the present study were added – see Table 3

      NM_000515.5: c.171delT;p.Phe57Leufs*43

      NM_005634.3:c.912G>A;p.Met304Ile

NM_173208.1 c.268C>T;p.Arg90Cys

NM_173208.1 c.82T>C;p.Ser28Pro

  1. - The variant has previously been identified (rs1555650923, with no MAF), reported in ClinVar database, and previously been described as de novo event in two unrelated families with HPE, type 4 (PMID: 11810641; PMID: 16962354). As the authors indicated, although the patient 3 shows clinical and instrumental features of HPE, it never has been associated to hypopituitarism. However, available clinical data in previously reported cases were not complete and we cannot exclude a priori that the growth deficiency and other features documented in those subjects cannot be associated also to hypopituitarism.

Answer: The TGIF1 p. Arg90Cys variant was first reported by Chen et al in 2002 (PMID: 11810641) in a fetus with HPE that was aborted at 22 wk.  The variant was de novo, and the fetal ultrasound showed a single cerebral ventricle with fused thalami consistent with alobar HPE, brachycephaly, hypotelorism, nasal hypoplasia and median orofacial cleft.  El-Jaick et al reported three novel variants in TGIF1 in patients with HPE and tested them functionally, along with some previously reported variants like p.Arg90Cys.  El-Jaick demonstrated that p.Arg90Cys causes a loss of function (PMID:16962354).  The ClinVar submission appears to be reporting the original case discovered by Chen et al.  and functionally tested by El-Jaick.   Thus, we believe that there is only one other case with TGIF1 p.Arg90Cys in the literature.  We have added a comment in line 323  to indicate that the pituitary status of the previous case was unknown.

  1. -The 268C>T;p.Arg90Cys variant has been found to be shared by other healthy members of the family (the healthy sister and father). This review agrees that clinical spectrum of malformations is consistent with the fact that the impact of TGIF1 variants may be variable and incomplete penetrance can also occur. In this context, a dedicated study on the primary transcript, or CpG islands within the promoter, may help to better understand this issue.

Answer: Thank you for your comment. There is a precedent for digenic or oligogenic disease in HPE, thus, WES of the family may identify variants in other genes that affect the penetrance of expression of the R90C allele (reference: Am J Hum Genet. 2002 Nov; 71(5): 1017–1032.

Published online 2002 Oct 22. doi: 10.1086/344412

PMCID: PMC385082

PMID: 12395298

Multiple Hits during Early Embryonic Development: Digenic Diseases and Holoprosencephaly

Jeffrey E. Ming and Maximilian Muenke).

  1. RESULTS, Patient 4 with allelic variant TGIF1 (c.82T>C;p.Ser28Pro). The variant has previously been identified (rs1171035105, MAF=1/125568, TOPMED). There is a large number of isoforms for this gene/protein, please provide the relative NM.. (or at least ENST..) and NP.. numbers. The variant is classified as VUS but deserves to be reported by this work, corroborating what has been previously reported on the affected Serine residue, thus providing clinical relevance for HPE. It could be interesting obtain a clinical comparison between the clinical profile of the subject carrying TGIF1(ENST00000330513.5):c.470C>G, p.Ser157Cys (rs121909066, PMID: 10835638) and the patient 4 of the present study.

Answer: Thank you for your comments. We added that information. See line 332-334 and 337-338.

  1. Table 3. Please insert the in silico predictions referring each variant (i.e. CADD and/or REVEL).

Answer: As requested, both predictors have been added to Table 3. However, REVEL did not return a result for the GH1 variant as it is a deletion.

  1. DISCUSSION: the authors should add the MIM reference numbers (i.e. Growth hormone deficiency, isolated, type IA, MIM#262400).

Answer: It was added in line 286

Overall, the manuscript provides evidence that combined pituitary hormone deficiency may be caused by different molecular events, that can be identified by high-throughput sequencing approaches. .

Reviewer 2 Report

Methods - it is not clear what version of the human genome assembly is used.

Variant chromosome position numbering. - consider using HGVS recommended format ( chrX:139586314:C>T, not X:139586314:C:T)

Title: Allelic variants in established hypopituitarism genes expands our knowledge of phenotypic spectrum. - consider changing to expand.

Only growth hormone deficiency (paragraph 3.2.1., page 5) - consider changing to isolated.

Alatzoglou et al., described the association of SOX3 with topical neu-rohypophysis and the craniopharyngeal channel persists, (paragraph 4.2, page 9). - the meaning is not clear, the phrase needs editing.

Author Response

Comments and Suggestions for Authors

Methods - it is not clear what version of the human genome assembly is used.

Answer: METHODS section 2.3, lines 67 to 79 has been changed to more clearly describe what was used for Panel and Exome sequencing. With that, the answer regarding human genome assembly has been added, which was hg19.

Variant chromosome position numbering. - consider using HGVS recommended format ( chrX:139586314:C>T, not X:139586314:C:T)

Answer: It has been changed in Table 3, as well as lines 107, 136, 156, 173, 207, 229, 250, 265.

Title: Allelic variants in established hypopituitarism genes expands our knowledge of phenotypic spectrum. - consider changing to expand.

We changed the title as recommended.

Only growth hormone deficiency (paragraph 3.2.1., page 5) - consider changing to isolated.

Answer: We took out the word only

Alatzoglou et al., described the association of SOX3 with topical neurohypophysis and the craniopharyngeal channel persists, (paragraph 4.2, page 9). - the meaning is not clear, the phrase needs editing.

Answer: We deleted this paragraph because it is not necessarily relevant to the patients we discuss here.

Round 2

Reviewer 1 Report

This is an improved version of the manuscript. Many reviewer's suggestions have been followed, thank you. 

There are still minor edits: 

  • TABLE3: please change "patogenic" with "pathogenic"
  • LINES 174-176: It is confusing and conflicting the concept that "the variant was not found in any of the population databases, .. " and that "It has been reported in gnomAD in two hemozygotes, ...". This phrase need be revised taking account that gnomAD is a browser based on several population studies.  
  • LINES 311-315: basing on the conflicting data about pathogenicity of the SOX3 change (p.Met304Ile), the clinical relevance of the variant remains still unclear. This should be add into the discussion section. In addition, it also needs to be indicated in the text that the molecular diagnosis has been reached out by sequencing of a target gene panel only and that WES or WGS approaches are required to exclude alternative molecular events underlying the disorder. 

Author Response

  • TABLE3: please change "patogenic" with "pathogenic"-
    • answer: we have corrected them accordingly 
  • LINES 174-176: It is confusing and conflicting the concept that "the variant was not found in any of the population databases, .. " and that "It has been reported in gnomAD in two hemozygotes, ...". This phrase need be revised taking account that gnomAD is a browser based on several population studies.
    • Answer:  We have deleted the part " the variant was not found in any of the population databases"  that in the present document was corrected to keep only " This variant has been reported in gnomAD in two hemizygotes, with a population frequency of 0.07e3."  in lines 171-172 highlighted in green.
  • LINES 311-315: basing on the conflicting data about pathogenicity of the SOX3 change (p.Met304Ile), the clinical relevance of the variant remains still unclear. This should be add into the discussion section. In addition, it also needs to be indicated in the text that the molecular diagnosis has been reached out by sequencing of a target gene panel only and that WES or WGS approaches are required to exclude alternative molecular events underlying the disorder. 
    • Answer: the lines 310-314 were changed to add your comments.
